# What Have We Learned from Patients Who Have Arboleda-Tham Syndrome Due to a De Novo *KAT6A* Pathogenic Variant with Impaired Histone Acetyltransferase Function? A Precise Clinical Description May Be Critical for Genetic Testing Approach and Final Diagnosis

**DOI:** 10.3390/genes14010165

**Published:** 2023-01-07

**Authors:** Nenad Bukvic, Massimiliano Chetta, Rosanna Bagnulo, Valentina Leotta, Antonino Pantaleo, Orazio Palumbo, Pietro Palumbo, Maria Oro, Maria Rivieccio, Nicola Laforgia, Marta De Rinaldis, Alessandra Rosati, Jennifer Kerkhof, Bekim Sadikovic, Nicoletta Resta

**Affiliations:** 1Medical Genetics Section, University Hospital Consortium Corporation Polyclinics of Bari, 70124 Bari, Italy; 2U.O.C. Genetica Medica e di Laboratorio, Ospedale Antonio Cardarelli, 80131 Napoli, Italy; 3Department of Biomedical Sciences and Human Oncology (DIMO), Division of Medical Genetics, University of Bari “Aldo Moro”, 70124 Bari, Italy; 4Division of Medical Genetics, Fondazione IRCCS Casa Sollievo della Sofferenza, 71013 San Giovanni Rotondo, Italy; 5Department of Biomedical Science and Human Oncology (DIMO), Section of Neonatology and Neonatal Intensive Care Unit, University of Bari “Aldo Moro”, 70124 Bari, Italy; 6Scientific Institute IRCCS “E. Medea”, Unit for Severe Disabilities in Developmental Age and Young Adults (Developmental Neurology and Neurorehabilitation), 72100 Brindisi, Italy; 7Department of Medicine, Surgery and Dentistry “Schola Medica Salernitana”, University of Salerno, 84081 Baronissi, Italy; 8Department of Pathology and Laboratory Medicine, Western University, London, ON N6A 3K7, Canada

**Keywords:** KAT6A, Arboleda-Tham syndrome, clinical exome sequencing, methylation studies, episignature, SNP array

## Abstract

Pathogenic variants in genes are involved in histone acetylation and deacetylation resulting in congenital anomalies, with most patients displaying a neurodevelopmental disorder and dysmorphism. Arboleda-Tham syndrome caused by pathogenic variants in KAT6A (Lysine Acetyltransferase 6A; OMIM 601408) has been recently described as a new neurodevelopmental disorder. Herein, we describe a patient characterized by complex phenotype subsequently diagnosed using the clinical exome sequencing (CES) with Arboleda-Tham syndrome (ARTHS; OMIM 616268). The analysis revealed the presence of de novo pathogenic variant in KAT6A gene, a nucleotide c.3385C>T substitution that introduces a premature termination codon (p.Arg1129*). The need for straight multidisciplinary collaboration and accurate clinical description findings (bowel obstruction/megacolon/intestinal malrotation) was emphasized, together with the utility of CES in establishing an etiological basis in clinical and genetical heterogeneous conditions. Therefore, considering the phenotypic characteristics, the condition’s rarity and the reviewed literature, we propose additional diagnostic criteria that could help in the development of future clinical diagnostic guidelines. This was possible thanks to objective examinations performed during the long follow-up period, which permitted scrupulous registration of phenotypic changes over time to further assess this rare disorder. Finally, given that different genetic syndromes are associated with distinct genomic DNA methylation patterns used for diagnostic testing and/or as biomarker of disease, a specific episignature for ARTHS has been identified.

## 1. Introduction 

KAT6A (Lysine Acetyltransferase 6A; OMIM 601408), KAT6B (Lysine Acetyltransferase 6B; OMIM 605880), KAT5 (Lysine Acetyltransferase 5; OMIM 601409) and KAT7 (Lysine Acetyltransferase 7; OMIM 609880) are part of the MYST family of proteins involved in a wide range of important cellular functions, such as chromatin remodeling, gene regulation, protein translation, metabolism and cellular replication [1]. 

Pathogenic variants (PV) in different genes that function as part of chromatin remodeling complexes have been discovered using an exome sequencing approach in some patients with syndromic intellectual disability [2,3,4] involving speech delay [5,6,7].

PV of KAT6A, just as those of KAT6B, have been associated with syndromic developmental delay, characterized by highly expressed phenotypic heterogeneity. On the one hand, *KAT6A* is linked to autosomal dominant intellectual disability, craniofacial-anomalies-cardiac-defects syndrome, recently described as Arboleda-Tham syndrome (OMIM 616268) [3,4]. On the other hand, KAT6B causes a spectrum of disorders, including genitopatellar syndrome (OMIM 606170), Ohdo syndrome (Say–Barber–Biesecker–Young–Simpson SBBYS variant OMIM 603736) and a Noonan syndrome-like disorder [8,9,10].

In addition, the KAT6A (NM_006766.4), also known as the MOZ or MYST3 gene, is recurrently rearranged or amplified in leukemia and non-hematologic malignancies, as well as involved in developmental disorders. Recently, sporadic or de novo heterozygous variants in KAT6A gene have been described to cause distinct intellectual disability syndrome with specific syndromic features (hypotonia, early feeding, motor difficulties, microcephaly and/or craniosynostosis, and cardiac defects) and subtle facial features [3,4]. The KAT6A gene is a histone acetyltransferase that causes chromatin modification and regulates a broad range of chemical processes in a wide range of conditions. Indeed, this gene has multiple functions in transcriptional regulation and developmental gene expression, as well as providing instructions for the production of proteins that are important for a variety of body functions. Here, we describe the case of a six-month-old infant sent to our genetic ambulatory counseling due to hypotonia, generalized dysmorphic characteristics, and a possible megacolon, throughout the diagnostic process.

Clinical exome sequencing (CES) revealed the presence of a de novo variant c.3385C>T (p.Arg1129*) in KAT6A (ACMG class IV PVS1, PM2) in the patient with this complex phenotype, which led to the diagnosis of Arboleda-Tham syndrome (ARTHS; OMIM 616268). Due to the dearth of cases described in the literature, our patients’ phenotypic data were compared to those previously documented.

Last but not least, we underline the need for straight multidisciplinary collaboration and precise clinical description findings, as well as the utility of CES in establishing an etiological basis in clinical and genetical heterogeneous conditions. Finally, as different genetic syndromes have been associated with unique genomic DNA methylation patterns that can be used for diagnostic testing and/or as biomarkers of disease, we tested episignature for ARTHS in our patient.

Therefore, we suggest additional diagnostic criteria that may help in the establishment of future clinical diagnostic guidelines, taking into account the reviewed literature, the phenotypic features characteristics of our patient, and the rarity of the disorder.

## 2. Materials and Methods

### 2.1. Patient Recruitment

The patient’s parents provided written informed consent to perform genetic testing and to publish clinical pictures together with the full content of this publication in accordance with the Declaration of Helsinki (1984) and its subsequent revisions, same as for any other applicable local ethical and legal requirements.

### 2.2. Clinical History

The medical records of the patient were reviewed.

The proband was the first male child of healthy non-consanguineous Brazilian parents. At the time of his birth, his father and mother were 29 and 28 years old, respectively. Family history was unremarkable, especially for neurodevelopmental disorders, brain abnormalities, recurrent miscarriages, other birth defects and/or genetic illnesses.

The child was born after an uneventful full-term gestation, at 39 weeks, through elective cesarean section due to podalic presentation. Birth weight was 2.950 g (>25th centile), length 50 cm (50th centile), head circumference 36 cm (85th centile) and APGAR scores 9/9 at 1′/5′, respectively. After birth, the patient was transferred to a Neonatal Intensive Care Unit (NICU) due to intestinal sub-obstruction and biliary vomiting on day 1. The extended neonatal screening for metabolic diseases (including Cystic Fibrosis) were negative. Renal ultrasonography showed a pyelectasis of 10 mm of the left kidney and a patent foramen ovale was observed when echocardiography was carried out. Due to persistent intestinal distention with difficult evacuations, Hirschsprung disease was suspected, but opaque clisma, conducted on day 13, was inconclusive. The patient was then sent home with enemas twice a day and close follow-up. At the age of ~6 months, a slight hypertonia of the limbs with closed fists, as well as non-visual coupling with alternate exotropia, were evident; head circumference was 42 cm (15th centile), weight 6.0 kg (>10th centile) and height 63 cm (50th centile). Genetic ambulatory counseling was started from ~6 months up to 36 months. Due to another episode of sub-occlusion, at the age of 7 months, colonic biopsies were performed but Hirschsprung disease was excluded. At the age of 27 months, the head circumference was 46.5 cm (10th centile), weight 12.0 kg (5th centile) and height 90 cm (50th centile), but the patient was unable to sit without assistance, could not mimic speech sounds, could not show grasping and could not focus on his hands. Truncal hypotonia with pronounced hypertonia of the limbs, strongly closed hands (fist) and flexion of the toes were observed. Dysmorphic facial features were evident: large forehead, bitemporal narrowing, high arched eyebrows, telecanthus, flat nasal bridge, low-set posteriorly rotated ears, thick earlobes, bulbous nose tip and thin upper lip, which later on became thick (Figure 1). Based on objective examinations during follow-ups from 6 to 36 months, hypothetical extensive differential diagnoses were made in the context of DD/ID, which lead to the execution of CES analysis.

### 2.3. Clinical Exome Sequencing (CES) as Previously Reported [11]

A clinical exome sequencing panel kit was used to perform next generation sequencing analysis on genomic DNA from peripheral venous blood (QIAamp DNA Blood Mini Kit, QUIAGEN Science, Germantown, Meryland, USA). More than 150,000 probes were designed based on human genome sequences to enrich approximately 11 Mb (114.405 exons) of conserved coding regions that cover >4900 genes (Sophia Genetics SA, Saint Sulpice, Switzerland). On the MiSeq instrument, library preparation and sequencing were carried out according to the manufacturer’s instructions (Illumina, San Diego, CA, USA). The average coverage depth was 70×. Raw data were analyzed using SOPHiA™ DDM (Sophia Genetics SA) with algorithms for alignment including single nucleotide polymorphisms (SNPs), insertions/deletions (Pepper™, Sophia Genetics SA patented algorithm), and copy number variations (Muskat™, Sophia Genetics SA patented algorithm). The raw reads were aligned to the human reference genome (GRCh37/hgl9), and the binary alignment map (BAM) files were visualized using an integrative genomics viewer (IGV).

### 2.4. SNP Array Analysis

The CytoScan HD array (Thermo Fisher Scientific, Waltham, MA, USA) was used to perform high resolution SNP array analysis of the proband and his parents, as previously described [12].

More than 2.6 million markers for copy number variations (CNVs) analysis and approximately 750,000 SNP probes capable of genotyping with a 99 percent accuracy are included in this array.

Thermo Fisher Scientific’s (Waltham, MA, USA) Chromosome Analysis Suite Software version 4.1 was used to analyze the data, which followed a standardized pipeline described in the literature [9]. The University of California Santa Cruz (UCSC) Genome Browser, build GRCh37, provided base pair positions, information about genomic regions and genes affected by CNVs, as well as known associated diseases (hg19).

### 2.5. Methylation Studies

Methylation analysis was performed with the clinically validated EpiSign^TM^ assay as previously described [13,14,15,16]. Briefly, methylated and unmethylated signal intensity generated from the EPIC array was imported into R 3.5.1 for normalization, background correction and filtering. Beta values ranging from 0 (no methylation) to 1 (complete methylation) were calculated as a measure of methylation level and processed through the established support vector machine (SVM) classification algorithm for EpiSign disorders. The EpiSign knowledge database composed of over 10,000 methylation profiles from reference disorder-specific and unaffected control cohorts was utilized by the classifier to generate disorder-specific methylation variant pathogenicity (MVP) scores. MVP scores are a measure of prediction confidence for each disorder, ranging from 0 (discordant) to 1 (highly concordant). A positive classification typically generates MVP scores greater than 0.5; in combination with assessment of hierarchical clustering and multidimensional scaling, these scores are used in generating the final matched EpiSign result.

## 3. Results

### 3.1. Genetic Findings

Trio clinical exome sequencing (CES) showed a de novo heterozygous, nonsense variant c.3385C>T in KAT6A gene (OMIM 601408): that result is a premature termination codon (PTC) (p.Arg1129*). No other variants were identified through exome sequencing. In addition, normal results were observed for karyotype and SNP array.

### 3.2. 3D Modelling

A KAT6A truncating protein of 1129aa resulting from de novo heterozygous variant (p.Arg1129*) was modeled using I-TASSER (Interactive Threading ASSEmbly Refinement https://zhanglab.ccmb.med.umich.edu/I-TASSER/ (accessed on 2 August 2021), a hierarchical approach used to predict protein 3D structure and function (pls. see Figure 2).

The ChemDraw programme was used to load and view *.pdb files created by I-TASSER. (version 8; Cambridge Software; PerkinElmer, Inc., Waltham, MA, USA).

### 3.3. Methylation Studies

EpiSign^TM^ is a clinically validated, genome-wide DNA methylation assay that enables detection of episignatures, epigenetic biomarkers that can be used for diagnostic screening and functional characterization of genetic variants in a growing number of genetic disorders. (PMID: 35047860; PMID: 33547396) EpiSign^TM^ screens for over 50 conditions and operates as a functional assessment for variant pathogenicity by comparing the DNA methylation profile of the patient with those from reference-affected and unaffected control cohorts to generate disorder-specific methylation variant pathogenicity (MVP) scores. EpiSign assessment of the patient revealed a genome-wide DNA methylation profile concordant with the methylation signature observed in patients with KAT6A variants, represented by an elevated MVP score of 0.27 (pls. see Figure 3).

## 4. Discussion

Arboleda-Tham syndrome is a rare genetic neurodevelopmental disorder characterized by global developmental delay (DD) and variable degrees of intellectual disability (ID) with limited or absent speech development associated with neonatal hypotonia, feeding difficulties, cardiac anomalies and dysmorphic facial features (predominantly broad nasal tip and thin, tented upper lip (ORPHA:457193)). In this syndrome, hypotonia contributes to motor delay, which is commonly observed [17]. Despite hypotonia not being reported in the patient’s medical documentation, further reviewed by us, truncal hypotonia associated with limb hypertonia was noted during our first visit (6 months). However, the patient’s Apgar score resulted in 9/9. This was also reported by Kennedy et al. [17], who described the largest cohort of patients in the literature. The combination of motor delay, specific speech/language delay and ID are universal [17] and, nowadays, our patient (~3yr) is nonverbal. Sometimes, verbal dyspraxia [17] is commonly associated with feeding difficulties (including nasogastric feeding), present in our patient too, due to oral motor dysfunction/dysphagia. The latter was pointed out by the mother, who referred to the mastication problem. Furthermore, our patient presented with reflux, constipation, bowel obstruction/intestinal malrotation, the clinical situation also reported with high prevalence between patients in Kennedy et al.’s [17] cohort. In our case, differential diagnosis and choice of appropriate genetic test was difficult considering the overlap in phenotype, which had to be taken into account due to some clinical suspicion discovered while reviewing the patient’s medical records. Namely, two different clinical entities characterized by genetic heterogeneity with high phenotypic overlapping such as Goldberg Shprintzen Megacolon Syndrome—GSMCS (*KIF1BP*)—associated with Hirschsprung disease, while ARTHS (KAT6A), as opposed to megacolon, is characterized by gastrointestinal obstruction/malrotation. The latter has been previously reported in KAT6B related disease [18]. Even though there is a direct relationship between syndromes and genes in ARTHS and GSMCS, it is known that multiple syndromes can be caused by pathogenic variants in the same gene (phenotypic heterogeneity); similarly, a single disorder can be caused by variants in the same gene or in different genes (genetic heterogeneity).

Cardiac malformations (including patent foramen ovale, observed in our patient) are present in 51% of Kennedy et al.’s cohort [17]. Regarding the hematological and immunological association described by Kennedy et al.’s cohort [17] and recurrent infections, our patient required surgery for obstruction and subsequently experienced sepsis. During hospitalization, few blood transfusions were required. Finally, facial characteristics (see Figure 1), intermittent strabismus, sleep disturbance, and brachydactyly were observed in our patient, which are consistent with those reported in Keneddy et al.’s study [17].

To avoid clinical bias and to support objectivity in differential diagnostics formulation, as well as to choose appropriate genetic testing, the Phenomizer has been adopted to perform searches based on sets of criteria (phenotypic abnormalities noted as diagnostic criterion—see Table 1). The Phenomizer is a web-based application for clinical diagnostics in human genetics using semantic similarity searches in ontologies’ Human Phenotype Ontology (HPO) [19,20].

The clinical exome sequencing (CES) was chosen based on the obtained results and other aspects (time, cost–benefits, etc.), including normal SNP array and karyotype results. According to TRIOS (parents and patient) CES, exome 17 has a de novo nonsense heterozygous variant that results in a premature termination codon (p.Arg1129*). In addition, Sanger sequencing confirmed the variant. This variant is found in the acid domain, which contains a high number of arginine residues [17] and causes the KAT6A protein to be truncated, leaving the HAT domain intact (Figure 4).

Moreover, to investigate the KAT6A truncating protein, the in silico 3D modeling was used by inserting a premature stop codon inside exon 17. This variant produces a protein with 1129 amino acids (Figure 2).

To date, other five de novo heterozygous truncating variants in the C-terminal transactivation domain of KAT6A have been reported. The loss of the PML (Acute Promyelocytic Leukemia Inducer; OMIM 102578) interaction domain, which extends from amino acid (aa) 1517 to 1741, as well as of two different RUNX1-2 domains (extending from 1517–1642 aa and 1913–1948 aa), required for RUNX1-2 protein interaction and activation, are caused by the deletion of the carboxy-terminal portion (Figure 2).

The deletion of C-terminal domains may increase the availability of PML in the cytoplasm. PML overexpression has been linked to human mesenchymal stem cell (hMSC) proliferation inhibition due to the activation of apoptosis and cell cycle arrest, as well as boosting hMSC osteoblast differentiation capacity. Mineralized matrix formation and alkaline phosphatase (ALP) activity were significantly increased in PML overexpressing hMSCs via time-dependent elevation of ALP mRNA. Similarly, it is possible to assume an increase in RUNX2 cellular availability. RUNX2 (also known as *CBFA1*) overexpression was reported in adipogenesis inhibition, and reduction in lipid droplet formation. Moreover, RUNX2 plays a fundamental role in the induction of chondrocyte hypertrophy and terminal maturation, stimulating the osteogenesis during endochondral bone formation and the competence of ADSCs as target cells for bone tissue engineering. The concomitant increase in the cellular availability of RUNX2 and PML could explain a dysregulation in osteogenesis with repercussions in the early fusion of cranial sutures and possible dysfunctions in normal brain development.

As reported by Kennedy at al. [17], significant clinical variability is observed for patients with KAT6A pathogenic variants and it was not possible to phenotypically group these patients since there are no unique and unifying features. However, the authors [17] reported phenotypic differences between early-truncating pathogenic variants (exon 1–15) and late-truncating pathogenic variants (exon 16–17). Namely, a bias of increased severity of developmental delay and increased frequency of microcephaly, neonatal hypotonia, gastrointestinal complications and congenital heart defects are associated with second type of pathogenic variants i.e., late-truncated (our patient). This observation suggested a potential role for nonsense-mediated decay (NMD) mechanisms, resulting in haploinsufficiency, when early-truncating pathogenic variants (exon 1–15) were observed, while those in exons 16 and 17 (late-truncated) would not result in NMD. In contrast to this, mRNA (messenger RNA) would result in a translated but dysfunctional protein that may have gain of function or dominant negative effects [17].

KAT6A was also considered to be an epigenetic modulator, having molecular effects that might be modified by environmental factors and background genetic variation. [17]. As is well-known, KAT6A is a part of multi-subunit complex with other proteins, which form a complex of acetylated lysine residues on histone H3 tails and subsequently promote a wide range of developmental programs [21,22,23].

However, further functional studies are required to better understand the effects of possible pathogenic variants on molecular processes that can be traced back to specific phenotypic characteristics. Follow-up of the patient, reported in Figure 1., which represents our patient in different periods of time, provided more significant insight into the range of phenotypic ARTHS expressivity and phenotypic changes during long and meticulous follow-up.

Therefore, it is evident that, through the use of a multidisciplinary approach involving molecular and clinical diagnosis reverse phenotyping approach (through Phenomizer), as well as a personalized diagnostic workflow and reviewed literature, it is possible to redefine and then identify previously unreported clinical signs of ARTHS useful for future definition of clinical diagnostic guidelines (pls. see Table 2).

However, due to some limitations in our study (observation based on one patient), as well as the small number of patients reported in literature, novel findings and phenotypic differences in phenotypic presentation must be cautiously taken into consideration given that they might not be directly related to the ARTH syndrome.

Finally, ongoing large-scale research on epigenetics is increasingly recognized as playing an important role during various pathophysiological processes, with current data providing evidence for Mendelian disorders of epigenetics machinery and chromatin modification, which may play an important role in different illnesses with widespread downstream epigenetic consequences/dysregulation. As reported in Levy et al.’s [15] study as well, syndromes caused by the same or by functionally related genes, such as KAT6A and KAT6B, might be difficult to distinguish using episignature, an obstacle which seems to be overcome for ARTHS. This was the reason behind the inclusion of this patient in the cohort of subjects for episignature/epivariation study in progress.

In conclusion, additional subjects and functional studies will be needed to further assess this rare disorder through meticulous recording of time-dependent phenotypic changes. Moreover, given the severity and magnitude of its multi-system detriments, we emphasize the importance of early diagnosis of ARTHS using CES while suggesting long follow-up, which may help spare patients from unnecessary diagnostic examinations and treatments.

EpiSign (DNA methylation) analysis of peripheral blood from a patient with a pathogenic nonsense KAT6A variant indicates that our case has a DNA methylation signature similar to that of subjects with confirmed ARTHS, and his episignature is distinct from those of the controls (Figure 3). This has the potential to advance testing in diagnostic settings of ARTHS.

## Figures and Tables

**Figure 1 genes-14-00165-f001:**
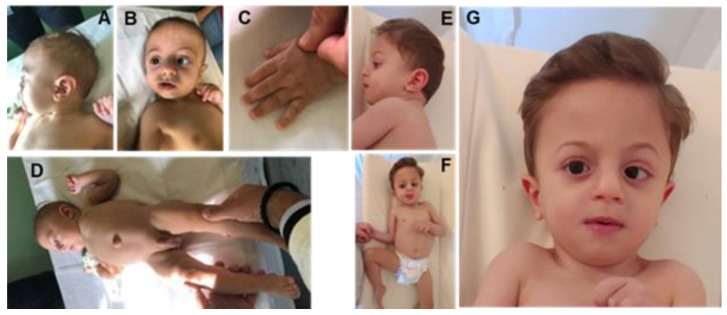
Patient at 6 months (**A**–**D**), 20 months (**E**–**G**).

**Figure 2 genes-14-00165-f002:**
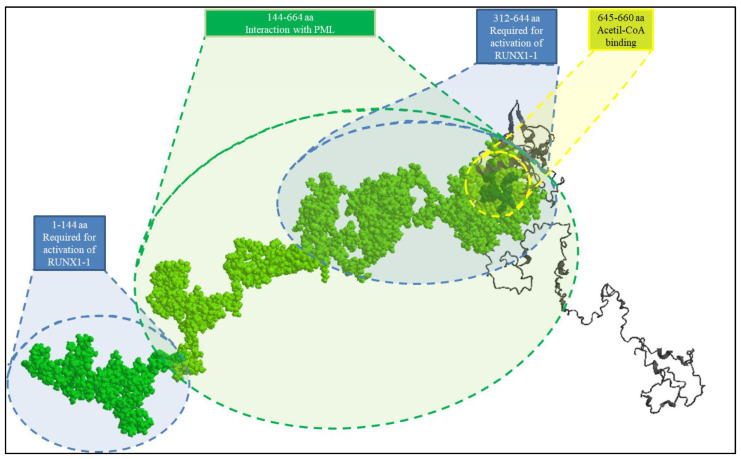
Three-dimensional structures of the 1129 amino acid (aa) KAT6A truncating protein. The PML interaction domain (aa 144–664) and the two RUNX1-2 domains (aa 1–144 and 312–644, respectively) are required for protein interaction and activation, which were highlighted in space-filling mode. The domain 645–660 aa of the acetyl-CoA binding site is marked with a yellow circle. The deletion of the C-terminal region may increase the availability of PML in the cytoplasm and the cellular distribution of RUNX2. The ability to recognize acetil-coA, on the other hand, stays unaltered.

**Figure 3 genes-14-00165-f003:**
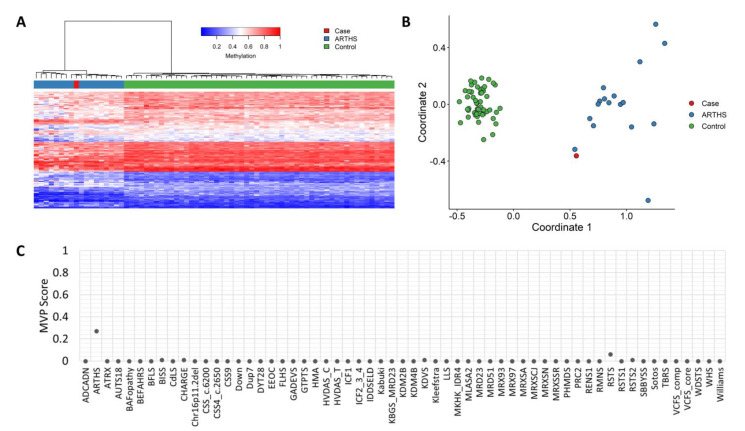
EpiSign (DNA methylation) analysis of peripheral blood from a patient with a pathogenic nonsense KAT6A variant, the causative gene for Arboleda-Tham syndrome 1. (**A**). Euclidean hierarchical clustering (heatmap); each column represents a single case or control, and each row represents one of the CpG probes selected for the episignature. (**B**). multidimensional scaling plot shows segregation of patient (red) has a DNA methylation signature similar to that of subjects with a confirmed ARTHS episignature (blue) and distinct from controls (green), confirming robustness of the episignature. (**C**). MVP score, a multi-class supervised classification system capable of discerning between multiple episignatures by generating a probability score for each episignature. The elevated patient score for ARTHS suggests an episignature similar to the ARTHS reference (see Methods section for methylation analysis).

**Figure 4 genes-14-00165-f004:**
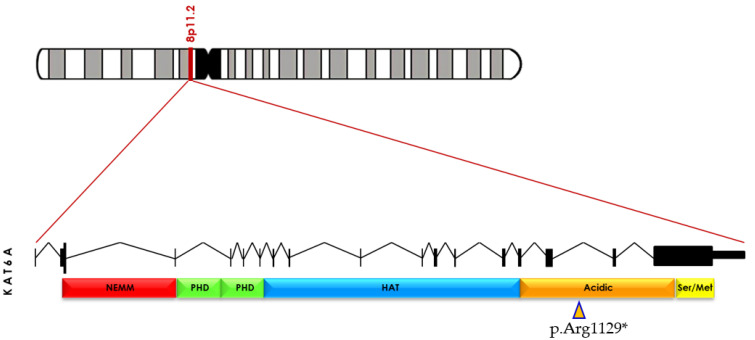
*KAT6A* (OMIM 601408) is located at 8p11.2. pathogenic variant c.3385C>T, resulting in a p.(Arg1129*), leading to the presence of premature termination codon (PTC) located in exon 17 (orange arrowhead). Truncated protein is missing part of acid (orange) and the serin/methionine-rich domains (yellow). Modified from Arboleda V. et al. [3].

**Table 1 genes-14-00165-t001:** ORPHA:457193 Autosomal dominant intellectual disability craniofacial-anomalies-cardiac defects syndrome. The phenotypic description of this disease is based on an analysis of the biomedical literature and uses the terms of the Human Phenotype Ontology (HPO). Phenotypic abnormalities are presented by order of frequency of occurrence in the patient population, then by alphabetical order inside each frequency group. * Phenotypic abnormalities noted as “diagnostic criterion” are those included in established sets of criteria to establish the diagnosis of a particular disease having been published in a peer-reviewed journal.

Diagnostic Criterion *
Patent ductus arteriosus HP:0001643
Abnormal facial shape HP:0001999
Intellectual disability, severe HP:0010864
Clinical signs and symptoms
Very frequent
Abnormal facial shape HP:0001999
Broad nasal tip HP:0000455
Global developmental delay HP:0001263
Intellectual disability, severe HP:0010864
Microcephaly HP:0000252
Narrow forehead HP:0000341
Neonatal hypotonia HP:0001319
Poor speech HP:0002465
Prominent nasal bridge HP:0000426
Thin upper lip vermilion HP:0000219
Frequent
Atrial septal defect HP:0001631
Cerebral visual impairment HP:0100704
Craniosynostosis HP:0001363
Downturned corners of mouth HP:0002714
Epicanthus HP:0000286
Feeding difficulties HP:0011968
Gastroesophageal reflux HP:0002020
Growth delay HP:0001510
Low-set, posteriorly rotated ears HP:0000368
Microretrognathia HP:0000308
Muscle stiffness HP:0003552
Neonatal respiratory distress HP:0002643
Patent ductus arteriosus HP:0001643
Plagiocephaly HP:0001357
Ptosis HP:0000508
Seizures HP:0001250
Short stature HP:0004322
Strabismus HP:0000486
Ventricular septal defect HP:0001629
Occasional
Brachydactyly HP:0001156
Cleft palate HP:0000175
Cryptorchidism HP:0000028
Dystonia HP:0001332
Hydronephrosis HP:0000126
Intestinal malrotation HP:0002566
Lacrimal duct stenosis HP:0007678
Laryngomalacia HP:0001601
Optic atrophy HP:0000648
Preauricular pit HP:0004467

**Table 2 genes-14-00165-t002:** Clinical signs and symptoms in order of HP number (modified from Phenomizer).

Clinical Signs and Symptoms
HP:0000219 Thin (thick in the early age) upper lip vermilion
HP:0000341 Bitemporal narrowing
HP:0000341 Narrow forehead
HP:0000368 Low-set posteriorly rotated ears
HP:0000368 Low-set, posteriorly rotated ears
HP:0000391 Thickened helices
HP:0000414 Bulbous nose
HP:0000431 Wide nasal bridge
HP:0000486 Strabismus
HP:0000286 Epicanthus/HP:0000506 Telecanthus
HP:0001156 Brachydactyly
HP:0001263 Global developmental delay
HP:0001319 Neonatal hypotonia
HP:0001510 Growth delay
HP:0001999 Abnormal facial shape
HP:0002003 Large forehead
HP:0002020 Gastroesophageal reflux
HP:0002509 Limb hypertonia
HP:0002553 Highly arched eyebrows
HP:0002566 Intestinal malrotation/stenosis of sigma-rectum junction
HP:0002465 Poor speech/HP:0001344 Absent speech
HP:0004322 Short stature
HP:0008936 Muscular hypotonia of the trunk
HP:0010864 Intellectual disability, severe
HP:0011968 Feeding difficulties

## Data Availability

The datasets used and/or analyzed during the current study are available from the corresponding author upon reasonable request for research only.

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
