# Peer review of "What Have We Learned from Patients Who Have Arboleda-Tham Syndrome Due to a De Novo KAT6A Pathogenic Variant with Impaired Histone Acetyltransferase Function? A Precise Clinical Description May Be Critical for Genetic Testing Approach and Final Diagnosis"

_genes, 2023, doi:10.3390/genes14010165_

Round 1

Reviewer 1 Report

This study is timely and much needed in the field of clinical/ medical genetics. Here, in addition to identifying a novel disease-causing variant in KAT6A that is associated with a recent syndrome neurodevelopmental disorder, the authors also highlight the need for better clinical phenotyping to help provide a clear diagnosis for patients with overlapping clinical features.

Further, the functional validation of this variant, particularly using clinical methylation testing such as EpiSign provides a resounding argument for better phenotypic characterization using standard of care clinical testing methods to help identify the genetic etiology of rare diseases.

Some minor comments:

1.     Few grammatical mistakes in the manuscript were noticed. A thorough proof reading can help fixing these accidental errors.

2.     The authors do not provide a clear description in the methods for Fig. 3 and 4. More elaboration for this will serve as a guide for other clinician scientists to incorporate such novel approaches in their diagnosis/ research.

3.     In the PCA plot in fig 4 (panel on right), you can observe that while the case clusters with other ARTHS cases harboring a KAT6A variant, within this group, there is some relative separation suggesting these samples have inter group variability. It would be interesting to see if the MVP scores of each sample was overlaid on this plot, whether the methylation patterns provide any insight into this and if it can be used to better finetune a diagnosis/ prognosis.

4.     Can the authors comment on how the epigenomic signature of KAT6A cases vary from KAT6B?

5.     Can the authors include in their discussion if there is any genotype-phenotype correlation observed in this condition thus far?

Reviewer 2 Report

This manuscript was a case report on an Arboleda-Tham syndrome patient with a complicated phenotype and a de novo mutation in KAT6A gene. The authors did an in-depth study on the case and analyzed it regarding the function of KAT6A as a histone acetyltransferase. They also reviewed the previous related studies and tried to present some additional points in this regard. The study was designed and presented well. However, There are a few points that considering them may improve the study: 

1. Since this is only a single case report study, novel findings and observations must be reported and interpreted with higher caution. Obviously, these findings might be not related to the ARTH syndrome. 

2. In the beginning of abstract and in entire manuscript, it is better to focus on the ARTH syndrome itself, instead of chromatin remodeling. 

3. It is recommended to add a section in "materials and methods" to briefly explain the process of clinical experiments by an expert clinician. 
